An ensemble learning based IDS using Voting rule: VEL-IDS

Emanet Sura
Karatas Baydogmus Gozde gkaratas@marmara.edu.tr
Demir Onder
Marmara University Istanbul , Istanbul , Turkey
Akleylek Sedat
Electronic publication date: 2023 Sep 29
Publication date: 2023
Volume: 9
Electronic Location ID: e1553
Received 2023 May 4; Accepted 2023 Aug 4
Copyright: ©2023 Emanet et al.
Copyright year: 2023
Copyright holder: Emanet et al.
License: This is an open access article distributed under the terms of the Creative Commons Attribution License, which permits unrestricted use, distribution, reproduction and adaptation in any medium and for any purpose provided that it is properly attributed. For attribution, the original author(s), title, publication source (PeerJ Computer Science) and either DOI or URL of the article must be cited.
License URL: https://creativecommons.org/licenses/by/4.0/

Keywords: Ensemble learning, Feature selection, Intrusion detection, Machine learning

Funding: The authors received no funding for this work.

==============================
Intrusion detection systems (IDSs) analyze internet activities and traffic to detect potential attacks, thereby safeguarding computer systems. In this study, researchers focused on developing an advanced IDS that achieves high accuracy through the application of feature selection and ensemble learning methods. The utilization of the CIC-CSE-IDS2018 dataset for training and testing purposes adds relevance to the study. The study comprised two key stages, each contributing to its significance. In the first stage, the researchers reduced the dataset through strategic feature selection and carefully selected algorithms for ensemble learning. This process optimizes the IDS’s performance by selecting the most informative features and leveraging the strengths of different classifiers. In the second stage, the ensemble learning approach was implemented, resulting in a powerful model that combines the benefits of multiple algorithms. The results of the study demonstrate its impact on improving attack detection and reducing detection time. By applying techniques such as Spearman’s correlation analysis, recursive feature elimination (RFE), and chi-square test methods, the researchers identified key features that enhance the IDS’s performance. Furthermore, the comparison of different classifiers showcased the effectiveness of models such as extra trees, decision trees, and logistic regression. These models not only achieved high accuracy rates but also considered the practical aspect of execution time. The study’s overall significance lies in its contribution to advancing IDS capabilities and improving computer security. By adopting an ensemble learning approach and carefully selecting features and classifiers, the researchers created a model that outperforms individual classifier approaches. This model, with its high accuracy rate, further validates the effectiveness of ensemble learning in enhancing IDS performance. The findings of this study have the potential to drive future developments in intrusion detection systems and have a tangible impact on ensuring robust computer security in various domains.

Introduction

Intrusion detection systems (IDSs) are generally used to detect anomalies which are widely used in network security infrastructure. IDSs can be divided into signature-based and anomaly-based in general. In the Signature-based IDSs, the characteristics of the attack are compared with the previously recorded attack patterns or signatures, and a match is searched. If a match is found as a result of the comparison, the activity is defined as an attack. This approach is insufficient in detecting new or unknown attacks. For this reason, it is used to detect more known attacks. In the anomaly-based IDSs, the basis of the attack detection process is based on the generation of different behaviors with deviations in the normal user behavior in the event of an attack (Mishra et al., 2018).

IDSs monitor network activity to detect policy violations or malicious activity. It analyzes the information obtained by monitoring and decides whether the activity is an attack or not. These systems, which are one of the important security components, are usually connected to a network Terminal Access Point or switch after the firewall and used in different ways. Figure 1 shows an IDS embedded in the network.

Figure 1 IDS in network.

Machine learning algorithms have been frequently used to improve the capability of IDSs to detect attacks due to their ability in learning. For this reason, researchers use these methods to increase the efficiency and performance of the systems with the aim of obtaining the lowest false alarm and highest accuracy rates (Kim, Aminanto & Tanuwidjaja, 2018).

In this study, voting ensemble learning model was developed that combines the benefits of selected machine learning algorithms to perform optimal intrusion detection and named as VEL-IDS. The advantage of ensemble learning over a single estimator is its ability to combine prediction results from several base classifiers to increase generalizability and robustness (Gao et al., 2019).

The efficiency of IDS is directly related to the quality of the dataset and the learning model. Many studies are based on datasets with known deficiencies such as anonymity (for privacy or ethical reasons), lack of traffic diversity, simulated traffic (not from a real-world network), and outdated attack traffic (Aljawarneh, Aldwairi & Yassein, 2018). In order to eliminate these shortcomings and contribute to the development of effective IDSs, the Canadian Cyber Security Institute, in cooperation with the Communications Security Organization, presented an updated IDS dataset named CSE-CIC-IDS2018 with real network traffic and high attack diversity.

This study focuses on improving intrusion detection performance by using ensemble learning and feature selection methods on the CSE-CIC-IDS2018 dataset. Various factors were considered in order to improve ensemble models. These are feature selection, base classifier selection, and ensemble learning method. In order to improve the accuracy rate and reduce the detection time of the intrusion detection model, the features determined from the CSE-CIC-IDS2018 dataset according to the results obtained by applying feature selection methods such as spearman’s correlation analysis, recursive feature selection (rfe) and chi-square test. The performance results were obtained and compared with the seven different machine learning methods such as adaboost, decision tree, logistic regression, multi-layer perceptron, extra trees, passive aggressive, and gradient boosting for the selection of the base classifier. Afterward, an ensemble model was created with the three classification methods in which the best results were obtained by applying the ensemble learnings ‘majority voting’ approach, which combines the benefits of each of the classification algorithms. The classifiers chosen for the ensemble model are logistic regression, decision tree, and extra trees.

Experimental results showed that when the RFE method was applied to the dataset by using the appropriate hyper-parameters for the algorithms selected for the ensemble learning model, run time is shortened and system performance increased. Accordingly, the ensemble model showed a better performance with an accuracy rate of 98.82% compared to individual approaches involving a single machine learning algorithm.

In the study Section 2 literature review, Section 3 Methodology, Section 4 Experimental results, Section 5 Discussion, and Section 6 Conclusion are given.

Related Work

In this section, information about studies in the literature is given. When these studies are examined, it is seen that generally old datasets are used for intrusion detection.

The intrusion detection performance of the IDSs may not always reliable because the models in AI algorithms generate a prediction for each incoming data. In 2020, Zhang, Li & Ye (2020) designed an IDS (BCNN-IDS) that can suppress unreliable detection output and improve overall detection performance to overcome the problem. NSL-KDD and UNSW-NB15 datasets were used for the study. Both binary and multiple classifications were done on these datasets. Bayesian-CNN and support vector machine (SVM) algorithms were selected for the AI algorithms used and an ensemble learning model was created. Evaluation results showed that the proposed model significantly improve accuracy and reduce false alarm rate by adopting an ensemble learning detection scheme.

Hsu et al. (2019) designed an intrusion detection system (ANIDS) based on a stacked ensemble learning model consisting of an auto-encoder (AE), SVM, and random forest (RF) algorithms. In order to show that the developed model is useful in different networks, NSL-KDD and UNSW-NB15 datasets were evaluated. They also observed the results by applying algorithms to the data in the campus network environment. Comparisons were done with three different machine learning models and result from two other related studies to evaluate the performance of the proposed model. Experimental results show that the proposed ensemble model provides a high classification accuracy.

In 2020, researchers designed a Natural Language Processing (NLP) and community machine learning-based intrusion detection system to detect anomalous traffic (Das et al., 2020). They used five machine learning algorithms such as logistic regression (LR), SVM, naive Bayes with Gaussian function (NB), decision tree (DT), and neural networks(NN). To do this, they vectorized the relevant information from HTTP requests by extracting natural language sentences and data preprocessing, and finally, they used the ensemble machine learning model to classify normal and abnormal traffic. Studies were conducted with HTTP DATASET CSIC 2010 to evaluate the performance of the proposed model. As a result of the experiments, it was seen that the accuracy rate of the proposed model was 99.96%.

Gautam & Doegar (2018) designed an ensemble learning model for binary classification that can achieve successful results on unbalanced datasets. In addition, one of the key points of the study is that they made a feature selection. KDD99 dataset, which is one of the oldest attack datasets, is preferred for the study. Three popular machine learning algorithms, NB, PART, and adaboost, were pioneers in model design. As a result of the study, it was seen that the average performance of the proposed Community Approach was better than the other classifiers.

In 2019, researchers designed a model that will detect DDoS attacks with the ensemble learning model (Das et al., 2019). To do this, feature selection is performed in the data preprocessing part. In the study, a subset was created by selecting features on the NSL-KDD dataset to detect DDoS attacks. Weka was used for the development. Among the machine learning methods, NN, SVM, k-nearest neighbor (KNN), and C4.5 were evaluated for the classification model. And at the end of all these, an ensemble learning model was designed. Comparisons with the existing papers showed that the proposed model had better detection accuracy with a lower false positive rate.

In 2019, researchers proposed an ensemble learning model that is a combination of long short-term memory (LSTM) and two ensembles (homogeneous and heterogeneous) methods (Adeyemo et al., 2019). The heterogeneous ensemble uses four standard classifiers with different computational properties (NB, KNN, RIPPER, and DT). Executions were performed on the UNSW-NB15 dataset in two formats: as a binary classification and as a multi-class classification. The proposed method had a detection accuracy of 97% and 85.23% in the binary dataset and the multi-attack classification.

Ravi, Chaganti & Alazab (2022) designed an ensemble learning model that performs Principal Component Analysis feature extraction and intrusion detection in the IoT. Experimental results of the proposed method on multiple comparative network intrusion datasets show that it outperforms existing methods and other most widely used machine learning and deep learning models. In particular, the proposed method showed a maximum accuracy of 99% in detecting network attacks and 97% in classifying network attacks using the SDN-IoT dataset.

In 2021, researchers developed an ensemble model to improve intrusion detection performance (Yousefnezhad, Hamidzadeh & Aliannejadi, 2021). SVM and KNN, which are popular machine learning algorithms, preferred to design an ensemble model. They also used the Dempster-Shafer method to evaluate more than one output. UNSW-NB15, CICIDS2017, and NSL-KDD datasets were selected for the study. Before classification with these datasets, feature selection is performed on them. As a result of the study, it was seen that the proposed method was more successful than basic machine learning models.

Jabbar, Aluvalu & Reddy (2017) designed a new ensemble model for intrusion detection system. The proposed approach efficiently classifies network traffic as normal or intrusion. RF and average one-dependence estimator (AODE) machine learning algorithms are preferred for the study and improvements done with the Kyoto dataset. The results show that the proposed classifier is more accurate than the NB, J48, and PART classifiers.

In 2019, researchers analyze the latest developments and recent issues in intrusion detection and propose an adaptive ensemble learning model (Gao et al., 2019). In the study, NSL-KDD dataset is preferred for training and testing processes. They used the MultiTree algorithm to balance the dataset. To improve the overall detection effect, they selected several base classifiers including DT, RF KNN, NN, and designed a voting ensemble learning algorithm. As a result of the study, they reached a success rate of 85.2%.

Table 1 shows the important parts of the related work.

Table 1 Comparison of the related work.

Paper	Year	Dataset	Algorithms	Feature selection extraction	Under sampling	Ensemble	Results	
Zhang, Li & Ye (2020)	2020	NSL-KDD
UNSW-NB15	CNN, SVM	No	No	Yes	Binary NSL-KDD 99.20
UNSW-NB15 98.50
Multi NSL-KDD 99.13
UNSW-NB15 97.59	
Hsu et al. (2019)	2019	NSL-KDD
UNSW-NB15	AE, SVM, RF	No	No	Yes	NSL-KDD 91.70
UNSW-NB15 91.90
Campus Data 96.10	
Das et al. (2020)	2020	HTTP DATASET CSIC 2010	LR, SVM, NB, DT, NN	No	No	Yes	99.96	
Gautam & Doegar (2018)	2018	KDD99	NB, Part, Adaboost	Yes	No	Yes	99.97	
Das et al. (2019)	2019	NSL-KDD	NN, SVM, KNN, C4.5	Yes	No	Yes	99.10	
Adeyemo et al. (2019)	2019	UNSW-NB15	LSTM, NB, KNN, RIPPER, and DT	No	No	Yes	Binary 97.00
Multi 85.23	
Ravi, Chaganti & Alazab (2022)	2022	SDN-IoT,
KDD99,
UNSW-NB15,
WSN-DS,
CICIDS-201	RNN, GRU, LSTM,
KPCA, RF, SVM	Yes	No	Yes	89.00 to 99.00	
Yousefnezhad, Hamidzadeh & Aliannejadi (2021)	2021	UNSW-NB15,
CICIDS2017,
NSL-KDD	KNN, SVM	Yes	No	Yes	UNSW-NB15 90.98,
CICIDS2017 98.97,
NSL-KDD 99.80	
Jabbar, Aluvalu & Reddy (2017)	2017	Kyoto	RF, AODE	No	No	Yes	90.51	
Gao et al. (2019)	2019	NSL-KDD	DT, RF KNN, NN	No	Yes	Yes	85.20	

Methodology

In this section, machine learning algorithms, feature selection methods, dataset, data preprocessing, ensemble learning, and the proposed new method in the study are explained.

Dataset

CSE-CIC-IDS2018 is a publicly available intrusion dataset developed in collaboration with the Canadian Institute of Cybersecurity and the Communications Security Organization (Sharafaldin, Lashkari & Ghorbani, 2018). This dataset was created by considering the deficiencies in previous intrusion datasets. CSE-CIC-IDS2018 is one of the biggest IDS dataset with real network traffic and a wide variety of attacks. It also contains normal and intrusion data. The intrusion data was created with real attacks implemented for 10 days using different attack tools. The dataset is organized daily, recording raw data for each machine including network traffic (PCAPs) and system logs. CICFlowMeter-V3, which is a network traffic flow generator and analyzer, is used in the feature extraction process from the raw data and 80 network traffic features saved as CSV files. When the dataset is received and analyzed on the local computer via Amazon Web Services, the number of data depending on the labels is shown in Fig. 2. In Fig. 2, attacks are shown in 14 different categories according to their type.

Figure 2 IDS in network.

There are six different attack types (2,748,235 attacks) in the dataset which are Botnet, DoS (Hulk, SlowHTTPTest, GoldenEye, Slowloris), Infiltration, BruteForce (Web, XSS, FTP, SSH), SQL Injection and DDoS(HOIC, LOIC-UDP, LOIC-HTTP). Figure 3 shows the distribution of attack types in the dataset.

Figure 3 Distribution of attack types in the dataset.

Data preprocessing

Before starting to work on the selected dataset, the data in 10 different .csv files were combined and several preprocessing operations were applied to them in order to work with feature selection and machine learning models. These preprocessing operations are performed according to the steps in publication (Karatas, Demir & Sahingoz, 2020). Data preprocessing steps are in the following part.

1. The Timestamp, Source IP, Flow ID, Destination IP, Destination Port, and Source Port features have been deleted from the dataset.

2. The NaN value is replaced by the 0 value.

3. The infinite values in the Flow Bytes/s and Flow Pkts/s columns are replaced by one more than the maximum value in the column.

4. InitBwd Win Bytes and InitFwd Win Byts columns can have a value of −1. Therefore, columns named InitBwdWinBytsNeg and InitFwdWinBytsNeg were created and assigned values of 1 or 0 considering the original attributes.

5. ‘Label’ gives the attack type information of the data. Therefore, the labels were numbered to represent the binary classification.

After the data preprocessing steps were completed, feature selection algorithms were performed. The results of the study on the machine learning models of the subset created by applying feature selection on the CSE-CIC-IDS2018 dataset are available in the publication named Emanet, Karatas Baydogmus & Demir (2021).

The dataset contains 16,232,943 data. A lot of preliminary work has been carried out to check whether it would be more efficient to work with a smaller version of this data instead of all of it. A reduction of 40%, 45%, and 50% applied to the size of the dataset and the results were examined. In order to simplify this large dataset and reduce the computation time, the dataset was under-sampled 50% by using the near-miss subsampling algorithm. The change in accuracy rate according to data size with the logistic regression algorithm, which is one of the fastest working algorithms, is shown in Fig. 4.

Figure 4 Underp-sampling result in logistic regression.

After the dataset was under-sampled by 50%, the number of data in the dataset downsized at 8,116,473. The attack distributions for each of the labels in the under-sampled dataset are shown in Fig. 5.

Figure 5 Number of labels in the under-sampled dataset.

Feature selection methods

Machine learning and data mining techniques are widely used to process and extract information from large-scale data. The fact that these methods are applied to large amounts of data containing irrelevant and unnecessary features affects the accuracy of the information and is costly in terms of time. In order to prevent this, popular feature selection algorithms are used in the literature and unnecessary features are eliminated. Since one of the aims of the study is to address the importance of feature selection in intrusion detection, three feature selection methods were evaluated for this study.

• Chi square test; It is based on whether the difference between the observed and expected frequencies is significant. In this method, it is tested whether there is a relationship between the features (X) and class Y. Based on the test result, features that are found to be unrelated to Y are removed from the dataset (Inc, 2007; Ünver & Gamgam, 2008; Budak, 2018).

• Correlation based feature selection (CFS) uses a search algorithm in addition to the function that measures the information values of feature subsets. The approach that CFS uses to measure the values of feature subsets considers the success of each attribute in estimating the class label and the internal correlation values between them. This approach is based on the hypothesis that good feature subsets consist of features that have a high correlation with the relevant class and low correlation with each other (Hall, 1999; Budak, 2018).

• Recursive feature elimination (RFE) is a common method for feature selection. The RFE method continuously removes the weakest features based on the iterative method and then ranks each feature in each iteration to delete the n lowest (score) features.

Feature selection on CSE-CIC-IDS2018 dataset

It is important for the performance of the developed IDS model that the dataset is cleaned of unimportant features and made to quality. For this reason, in order to determine the most appropriate subset from the CSE-CIC-IDS2018 dataset to be used in the study, analyses were made with the three feature selection approaches given earlier.

1. The score of each feature was calculated and the features with low scores were removed from the dataset with the Chi-square test.

2. High correlation features were determined by Spearman correlation analysis and removed from the dataset.

3. All the features ranked and the most irrelevant features removed from the dataset using RFE.

A comprehensive study was carried out beforehand to determine the feature selection method and feature numbers to be used in the study. Details of this study can be found in the Emanet, Karatas Baydogmus & Demir (2021) publication.

As a result of detailed execution, a different number of features and accuracy rates are achieved in each feature selection algorithm. In this manner, it has been seen that the selection of 31 features with Chi-Square, 25 features with Spearman correlation, and 40 features with RFE increases the accuracy rates. Table 2 shows the accuracy rates as a result of training without feature selection and with feature selection algorithms.

Table 2 Accuracy rates of the algorithms.

Model	Without selection	Chi Square	Spearman	RFE	
ADA	97.73	96.99	97.20	97.77	
DT	98.65	98.33	98.42	98.65	
ET	98.74	98.61	98.69	98.76	
GB	98.70	98.33	98.35	98.71	
LR	95.15	84.30	81.27	95.15	
MLP	97.66	96.41	95.94	97.77	
PA	91.82	77.36	85.07	94.77	

When Table 2 is examined, it is seen that RFE used with appropriate parameters in machine learning-based IDSs can increase accuracy. For this reason, the proposed work continued with the new dataset consisting of the features selected by the RFE method and an ensemble model created using this dataset.

Machine learning algorithms

Seven popular supervised machine learning algorithms were used in the study. These algorithms were selected among the most popular ones by searching the literature. Decision tree (Han, Kamber & Mining, 2006), extra trees (Geurts, Ernst & Wehenkel, 2006), logistic regression (Bayazit, Sahingoz & Dogan, 2020; Alpar, 2017), gradient boosting (Bentéjac, Csörgo & Martinez-Munoz, 2021), passive aggressive classifier (Gupta & Meel, 2021), and multi-layer perceptron (Ayşe & Berberler, 2017) algorithms were used in the study. The reason for selecting these machine learning algorithms is their popularity in the literature and their widespread preference in the field of intrusion detection. By choosing well-established algorithms, researchers who are familiar with these methods can readily apply our proposed approach. Furthermore, the selection of parameters for the study was guided by the (Karatas, Demir & Sahingoz, 2020) study, and we followed the suggested parameter settings outlined in this study.

Ensemble learning

Ensemble learning proposed by Nilsson for the classifiers in supervised learning which mostly outperform the models created using a single classifier is among the effective approaches used in machine learning (Subasi, 2020; Bilgin, 2018). In ensemble methods, the capabilities of individual classifiers which are called base classifiers, are brought together and it is aimed to improve the classification accuracy. Bagging, Boosting, Stacking, and Voting are the most well-known ensemble learning methods. In this study, the Voting method of ensemble learning was used and only information about it was given in the following.

Voting; is one of the easiest methods of combining predictions from many classifiers. In this method, different types of classifier groups are trained in parallel and combined in order to benefit from the characteristics of each of the classifiers (Akman, Genç & Ankarali, 2011). The classification process for different types of classifiers has a positive effect on increasing the accuracy rate. For this reason, providing classifier diversity will increase the accuracy rate (Polikar, 2012).

The most common unification rule in the voting approach is majority vote (Polikar, 2012). Mean and weighted average rules are used for classifiers that produce continuous results (Zhou, 2012). Figure 6 shows the voting steps of the collective learning algorithm.

Figure 6 Majority voting example.

Proposed system

IDSs should fulfill the demands and increasing needs in developing technology (Thomas & Pavithran, 2018). Machine learning approaches, which are preferred in many studies, are also used in this field (Athmaja, Hanumanthappa & Kavitha, 2017). The purpose of using these approaches in IDSs is to perform classification with high performance using the data that the system did not know before (Sahingoz et al., 2019). IDSs usually handle large-scale data which contains various redundant features that end in a low accuracy rate and long processing time (Amrita, 2013). This makes feature selection an important issue. Feature selection means choosing the most important features from the dataset in order to reduce the classification training time and increase the accuracy rate (Zhou et al., 2020). In this study, various feature selection and ensemble learning methods were examined in order to create an effective IDS with high accuracy in binary classification. Figure 7 shows the proposed model developed for IDS. The proposed model consists of three parts: data preprocessing, training/testing, and intrusion detection.

Figure 7 Proposed model.

In the data preprocessing, the steps described under the ’Data preprocessing’ section are carried out. The operations are briefly mentioned in the following;

• Detection/cleaning of inconsistencies,

• Correction incorrect data,

• Completing missing values,

• Scaling and normalization.

• Under-sampling

• Feature selection

There are two leading steps for the study; data under-sampling and feature selection. For under-sampling, size reduction was made on the data, and the results were observed as mentioned under the section ’Data preprocessing’. Accordingly, the size of the selected dataset was reduced by 40%, 45%, and 50%, and the results of the existing algorithms were evaluated. In addition, the dataset was under-sampled to 50% by keeping the data distributions the same as a result of the preprocessing in the dataset. Then in the preprocessing, the most important features in the dataset were selected using feature selection techniques. The determined feature selection techniques were applied separately and the results were observed. Time and other metrics are evaluated along with accuracy for results. Detailed information on this subject is given under the section ‘Feature Selection, on CSE-CIC-IDS2018 Dataset. As a result of the executions, it has been seen that the most suitable method for feature selection is RFE and a dataset with 40 features can increase the performance. In the training, an ensemble model was created by comparing the seven basic classifiers for each of the feature selection methods and combining the three main classifiers with the highest performance for the feature selection method that gives the best efficiency. In this part, the Stratified 5-Fold Cross-Validation technique is used to evaluate the performance of decision tree, gradient boosting, adaptive boosting, logistic regression, passive-aggressive, extra trees, and multi-layer perceptron classifiers. The results were evaluated according to the performance metrics of accuracy, recall, precision, F1-score, and calculation time (Karatas, Demir & Sahingoz, 2020; Sokolova & Lapalme, 2009).

In the last stage of attack detection, an evaluation of the ensemble model, whose performance was examined with the cross-validation technique was made.

Experimental Results

Ensemble learning is a machine learning method in which several learning algorithms are combined to design models that can increase accuracy (Polikar, 2012). There are four different types of ensemble learning techniques as Bagging, Boosting, Stacking, and Voting. In voting, the power of several single classifiers eases the application of a combination rule for decisions (More & Gaikwad, 2016). Since this study focuses on binary classification, the majority vote was chosen as the ensemble learning technique. Because the majority vote embraces democratic rules, means it depends on the result of the class that gets the most votes from the execution (More & Gaikwad, 2016). Each classifier generates its own prediction, and the model selects the highest number of prediction results produced in each classifier. In the study, a subset of the CSE-CIC-IDS2018 dataset with 40 attributes was created using the RFE and used for the ensemble model development. The parameters of all classifiers have been implemented as the default value in the Python scikit-learn library. Table 3 shows the accuracy rates and times obtained by machine learning algorithms using the original dataset.

Table 3 Results without under sampling and feature selection.

Model	Accuracy (%)	Error Rate (%)	Time (min.)	
ADA	97.73	2.27	35,36	
DT	98.65	1.35	14,04	
ET	98.74	1.26	20,53	
GB	98.70	1.30	157,55	
LR	95.15	4.85	3,09	
MLP	97.66	2.34	270,38	
PA	91.82	8.18	1,58	

The results show that tree-based algorithms are quite successful when the table is examined. Since the aim of the study is to design an ensemble model with the feature-selected dataset, in the next step all algorithms run with the RFE applied dataset. The results are shown in Table 4.

Table 4 Results using RFE without under-sampling.

Model	Accuracy (%)	Error (%)	Precision (%)	Recall (%)	F1-Score (%)	Time (min.)	
ADA	97.77	2.23	94.49	96.58	95.50	24,24	
DT	98.65	1.35	97.28	97.40	97.34	6,34	
LR	95.15	4.85	92.41	89.35	90.79	1,20	
MLP	97.77	2.23	95.80	95.51	95.68	177,19	
ET	98.76	1.24	97.21	97.86	97.53	8,19	
PA	94.77	5.23	86.19	87.03	79.62	0,35	
GB	98.71	1.29	96.34	98.52	97.39	117,04	

Figure 8 shows the model performances based on accuracy, precision, recall, f1-score, and time results reached using the RFE method.

Figure 8 Performance metrics with RFE.

Gradient boosting, adaboost, and multi-layer perceptron algorithms have high accuracy rates. However, the prediction times in these algorithms are quite long compared to other ones. In the Passive-Aggressive algorithm, the prediction time is quite short but the accuracy rate is low. When the prediction time and accuracy rate are evaluated together, it has been seen that the decision tree, logistic regression, and extra trees algorithms give results with high accuracy and fast prediction times. The accuracy rates and run times of these algorithms as a result of working with the original dataset and RFE applied datasets are given in Table 5.

Table 5 Results using RFE with under-sampling.

	Accuracy (%)	Time (min)	
Model	Original	With RFE	Original	With RFE	
DT	98.65	98.65	14.04	6.34	
ET	98.74	98.76	20.53	8.19	
LR	95.15	95.15	3.09	1.17	

Therefore, these three algorithms were selected for the ensemble model. Table 6 shows the execution results for the ensemble model.

Table 6 Results of the ensemble model.

Ensemble Model	Accuracy (%)	Error (%)	Precision (%)	Recal (%)	F1-S (%)	Time (min)	
40-Feature Dataset	98.82	1.18	98.23	97.09	97.65	6,42	
All Features	98.79	1.21	98.15	97.59	97.60	24,02	

It is seen that the error rate of the feature extracted dataset is 1.18% and the error rate of the original dataset is 1.21% when the table is examined. It means, the proposed new ensemble model reduced the error rate. Figure 9 shows the results of the performance metrics reached with the ensemble model.

Figure 9 All features vs 40-features.

It is clear that the proposed model is more successful in performance metrics when the Fig. 9 is examined. The accuracy of the proposed featured selected ensemble model is 98.82% which has a comparatively lower prediction time than using all features in the dataset. Specifically, the proposed model reduced the run time from 24 min 2 s to 6 min 42 s. The ROC/AUC ratios of the algorithms used in the study are shown in Fig. 10.

Figure 10 ROS/AUC ratios for all algorithms.

Discussion

In this study, it has been seen that using feature selection and ensemble learning methods can improve performance in machine learning-based IDSs. CSE-CIC-IDS2018 dataset was used in the implementation of the study.

In the first phase, under-sampling is done by using spearman’s correlation analysis, recursive feature elimination (RFE), and chi-square Test methods. The main purpose of feature selection is to improve intrusion detection performance by removing unnecessary and unimportant features. In order to see the effect of the used feature selection methods on intrusion detection performance, experiments were carried out on subsets of the CSE-CIC-IDS2018 dataset created with the features determined by each method. Decision tree, gradient boosting, adaboost, logistic regression, passive-aggressive, extra trees, and multi-layer perceptron classifiers were used to compare the subsets with the original size dataset. In all experiments, the Stratified 5-Fold Cross-Validation technique was used to evaluate the performance. GPU parallelization has been implemented to reduce the time and computational costs associated with this technique. The results obtained with spearman’s correlation analysis and chi-square test showed that tree-based models can detect attacks with a performance of over 97%. When tree-based model performances were evaluated according to the run time and accuracy rate, it was seen that spearman’s correlation analysis gave a superior performance than the chi-square test. When the results of spearman’s correlation analysis and chi-square test methods were compared with the results obtained using the original dataset, it was seen that both methods had a positive effect in terms of run time, but did not provide any increase in accuracy rates. The RFE method increased system performance in all models for both run time and accuracy rate. It would be insufficient to evaluate only the accuracy rate in a dataset that does not have a balanced distribution such as CSE-CIC-IDS2018, recall, precision, and F1-score results also examined. Experimental results showed that Spearman’s correlation coefficient and chi-square test methods shortened the run time by 45% due to the under-sampled dataset, but increased the error rate by 10.52% and 14.46%. In addition, the RFE method provided a 38% reduction in the run time and reduced the error rate of the system down to 2.95% when appropriate parameters were used. Therefore, performance improvement in machine learning-based IDSs can be achieved by using the RFE method with the correct hyperparameters.

In the second phase of the study, an ensemble model has been produced that benefits from combined classifiers. This model was created with the selected classifiers based on the performance results obtained in the first phase. For the performance evaluations of the ensemble model, a dataset with 40 attributes was used. Since the decision tree, logistic regression, and extra trees algorithms gave high-accuracy results with fast prediction times when the run time and accuracy rates were evaluated, these three models were used to create the ensemble model. The proposed ensemble model has a relatively low run time and a high accuracy rate of 98.82%, resulting in the use of all features. Specifically, the proposed model provided an approximate 73% reduction in detection time and a 3% increase in accuracy. Since the existing studies in the literature used NSL-KDD, KDD Cup99, UNSW-NB15, etc., it is not correct to compare them with our study. Since these datasets contain old and duplicate data, they do not provide up-to-date and realistic results like CSE-CIC-IDS2018.

Considering what has been done in the study, the advantages of it compared to the existing studies in the literature and the information about why it should be used are in the following section;

• Since Big Data is a part of our lives, algorithms are needed to perform attack detection by working efficiently and quickly with big data. The model we proposed uses powerful machine learning algorithms by processing big data and performs attack detection with high performance.

• Through to ensemble learning, the strengths of the selected machine learning algorithms were highlighted and the newly proposed model performed the operations with the strongest features of the selected algorithms.

• In the study, various feature selection methods and different machine learning algorithms were examined. In this direction, there is diversity in the study.

• The proposed model is open to development and designed to be used easily with different methods.

• In addition, when the studies in the literature are evaluated, the model proposed in this study has the advantages in the following part;

• KDD-Cup99 and NSL-KDD datasets are generally used in the literature. In this study, the CSE-CIC-IDS2018 dataset, which contains current and big data, was used. In this way, the results achieved are more realistic.

• Studies in the literature have generally examined a few machine learning algorithms or deep learning algorithms. The proposed model has comprehensively evaluated 8 different machine learning algorithms.

• Models that make feature selection in the literature generally choose a single method and work on that method. The proposed model examined three different feature selections and also obtained results with different feature numbers. This allowed the collective model to be more successful.

Although the aim of the study is to reduce the size of the dataset using statistical feature selection methods and design a collective model, there are also valuable studies in the literature that have employed swarm algorithms for feature selection. Researchers conducted a study in 2022 to investigate the effect of swarm algorithms on feature selection. According to this an enhanced evaluation metric called RF-measure is introduced to assess the impact of missing data on the performance of feature selection in the presence of class imbalance (Zhang et al., 2021). To enhance the performance of proposed model, two problem-specific strategies are devised: a swarm initialization strategy guided by fuzzy clustering and a local pruning operator based on feature importance. Experimental results, comparing proposed model with state-of-the-art feature selection algorithms across various public datasets, demonstrate its outstanding classification performance while requiring relatively less computation time. Another important study was carried out in 2023. In this study, a novel approach is employed to divide the entire sample set into multiple smaller sample subsets using a nonrepetitive uniform sampling strategy (Song et al., 2022). Each of these sample subsets is treated as a surrogate unit. Subsequently, a collaborative feature clustering mechanism is introduced to partition the feature space, thereby reducing the computational cost of feature clustering and narrowing down the search space for Particle Swarm Optimization (PSO). Building upon this, an ensemble surrogate-assisted integer PSO method is proposed. To ensure accurate prediction, an ensemble surrogate construction and management strategy is devised for evaluating particles. The feature selection methods in these studies will serve as inspiration for our future research.

While the study makes a valuable contribution to the existing body of research, it is important to acknowledge its limitations. These limitations, along with potential solutions, are outlined in the following section:

• In this study, the primary focus is on working with large-scale datasets. However, one of the major challenges encountered is the scarcity of real-world attack data for effective attack detection. The available datasets either rely on artificially generated data or contain substantial gaps in their information. To address this limitation, it is crucial to explore alternative avenues for acquiring authentic attack data. Collaborating with organizations or security agencies that can provide access to anonymized real-world attack datasets could offer a more realistic foundation. Additionally, efforts can be made to augment existing datasets by incorporating additional diverse and representative attack scenarios.

• Another objective of the study is to investigate the impact of the number of features in the educational context. However, finding datasets with a wide range of features poses a significant challenge. Intrusion detection datasets commonly exhibit similar sets of features, thus limiting the exploration of datasets with extensive feature sets. To overcome this limitation, researchers should consider expanding their scope beyond intrusion detection datasets. Exploring domains or industries outside of the traditional context may offer access to datasets that encompass a greater variety of features. For instance, datasets from domains such as finance, healthcare, or social media could provide a broader range of features to study.

• Furthermore, conducting the study necessitates a robust working environment capable of efficiently handling big data and executing 5-Fold cross-validation in parallel. Systems with limited GPU resources may encounter difficulties when running the model. To address this concern, it is advisable to utilize powerful computing systems equipped with sufficient GPU resources. Alternatively, leveraging cloud-based computing services that offer scalable resources can ensure smooth execution of the model on large-scale datasets while facilitating seamless 5-Fold cross-validation.

By acknowledging and addressing these limitations through the proposed solutions, future research in this field can enhance the validity and applicability of the study’s findings.

Conclusion

In the study, a new machine learning-based IDS model was proposed with the ensemble learning method by under-sampling and feature selection on large datasets. In this manner, CSE-CIC-IDS2018 has been used as the dataset to be studied for developments, which has been popularly used in recent years. The near-miss algorithm was used to perform under-sampling on the related dataset. Spearman’s correlation analysis, recursive feature elimination (rfe), and chi-square test methods were used for feature selection. Seven popular machine learning algorithms selected in the study, these are; adaboost, extra trees, multi-layer perceptron, decision tree, random forest, logistic regression, passive-aggressive, and gradient boost classification algorithms. All experiments were performed with Stratified 5-fold cross-validation. In the study, the size of the dataset was reduced to 40%, 45%, and 50%, and then intrusion detection is done using seven selected machine learning algorithms. Data under-sampling by 50% preferred for feature selection. The most successful algorithm in feature selection was RFE. Various performance metrics are used for the performance evaluation of the ensemble model. Considering the performance metrics,decision tree, logistic regression, and extra trees algorithms were chosen to create the collective model because they gave better results with high accuracy and fast run time. The proposed ensemble model has an accuracy rate of 98.82% with under-sampling and feature selection. In addition, the proposed model provided a 73% reduction in intrusion detection time and 3% increase in accuracy. The study will be a pioneer for researchers who will both work with the relevant dataset and seek different approaches.

Supplemental Information

Supplemental Information 1 Feature Selection Algorithm

Click here for additional data file.

Supplemental Information 2 Feature Selection Algorithm

Click here for additional data file.

Supplemental Information 3 Model Evaluation

Click here for additional data file.

Supplemental Information 4 Feature Selection Algorithm

Click here for additional data file.

Supplemental Information 5 Model Evaluation

Click here for additional data file.

Supplemental Information 6 Model Evaluation

Click here for additional data file.

Supplemental Information 7 Model Ealuation

Click here for additional data file.

Supplemental Information 8 Feature Selection Algorithm

Click here for additional data file.

Additional Information and Declarations

Competing Interests

Author Contributions

Data Availability

The authors declare there are no competing interests.

Sura Emanet conceived and designed the experiments, performed the experiments, analyzed the data, performed the computation work, prepared figures and/or tables, authored or reviewed drafts of the article, and approved the final draft.

Gozde Karatas Baydogmus conceived and designed the experiments, performed the experiments, analyzed the data, performed the computation work, prepared figures and/or tables, authored or reviewed drafts of the article, and approved the final draft.

Onder Demir conceived and designed the experiments, performed the experiments, analyzed the data, performed the computation work, prepared figures and/or tables, authored or reviewed drafts of the article, and approved the final draft.

The following information was supplied regarding data availability:

The base codes are available in the Supplemental Files.

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
