# Peer review of "An ensemble learning based IDS using Voting rule: VEL-IDS"

_PeerJ Computer Science, doi:10.7717/peerj-cs.1553_

## Round 0.1 · original submission · Major Revisions

This paper proposed a new IDS model with the ensemble learning method. According to the reviewers' comments, this article is quite innovative, but there are many problems and it still needs major revision.

Reviewer 1 ·

Basic reporting

This paper proposed a new machine learning-based IDS model with the ensemble learning method by under-sampling and feature selection on large datasets. Experimental results show that the proposed ensemble model has an accuracy rate of 98.82% with under-sampling and feature selection. According to their expression, writing and simulations, I think there still have some problems in the paper.
1> The abstract in current form is kind of superficial, and the abstract needs to rewrite to point out significance and impact of the paper.
2> The motivation description in this paper is not clear enough, the authors are requested to analyze the motivation mainly from the shortcomings of the current work. In addition, for the related work in Section II, this part should not simply list the research status of the existing work, the author should also discuss their advantages and disadvantages.
3> Some representative feature selection methods should be mentioned or used, such as self-adaptive particle swarm optimization for large-scale feature selection in classification, a fast hybrid feature selection based on correlation-guided clustering and particle swarm optimization for high-dimensional data.
4> Three feature selection methods were evaluated for this study. Why choose these feature selection algorithms? The author should provide the reasons for this.
5> 31 features with 225 Chi-Square, 25 features with Spearman correlation, and 40 features with RFE are selected respectively. Why not other numbers?
6> The pseudo-code of the proposed main algorithm must be presented for better understanding of readers.
7> Section 3.3 Machine Learning Algorithms is too simple. Why chooses these algorithms? How to use these algorithms? How to set their parameters? The author did not provide a corresponding explanation.

Experimental design

8> The results, especially the comparisons between the proposed algorithm, should be discussed more detailed. What are the insights? Why the proposed strategy/mechanism can achieve good results?

Validity of the findings

--

Additional comments

--

Cite this review as

Reviewer 2 ·

Basic reporting

1. To eliminate unnecessary features, the authors used feature selection methods. In addition to several types of methods mentioned in this paper, there are already some evolutionary algorithm-based feature selection methods with superior performance. For example, " Surrogate sample-assisted particle swarm optimization for feature selection on high-dimensional data, IEEE Transactions on Evolutionary Computation, DOI: 10.1109/TEVC.2022.3175226", "Clustering-guided particle swarm feature selection algorithm for high-dimensional imbalanced data with missing values, IEEE Transactions on Evolutionary Computation, DOI:10.1109/TEVC.2021.3106975 ", and so on.

Experimental design

1. Did the authors divide the whole dataset into two totally separate datasets, one only for training, and the other only for the final testing? Note that this is not just using cross validation.
2. Results of the experiments need to be statistically analyzed.

Validity of the findings

1. The limitations of the proposed method need to be discussed. Some of these limitations and their resolution should be collected into a discussion of future work.

Additional comments

1. How about the computation complexity of proposed method? Authors should give related explanation.

Cite this review as

---

## Round 0.2 · accepted · Accept

We are happy to inform you that your manuscript has been accepted for publication since the reviewers have been satisfied with the revision.

Reviewer 1 ·

Basic reporting

Authors have addressed all the issues according to my previous comments. The related work has been enriched and the indistinct description as well as deficient analysis has been further refined. More discussions have also been added. This paper has been revised thoroughly to reach the standard for publication. Consequently, I advise to accept this paper.

Experimental design

Authors have addressed all the issues according to my previous comments. The related work has been enriched and the indistinct description as well as deficient analysis has been further refined. More discussions have also been added. This paper has been revised thoroughly to reach the standard for publication. Consequently, I advise to accept this paper.

Validity of the findings

Authors have addressed all the issues according to my previous comments. The related work has been enriched and the indistinct description as well as deficient analysis has been further refined. More discussions have also been added. This paper has been revised thoroughly to reach the standard for publication. Consequently, I advise to accept this paper.

Additional comments

Authors have addressed all the issues according to my previous comments. The related work has been enriched and the indistinct description as well as deficient analysis has been further refined. More discussions have also been added. This paper has been revised thoroughly to reach the standard for publication. Consequently, I advise to accept this paper.

Cite this review as

Reviewer 3 ·

Basic reporting

It is much improved from the previous version.

Experimental design

no comment

Validity of the findings

no comment

Additional comments

Use consistent time unit expressions (min. OR min) in both tables and figures.

Cite this review as